# Machine Learning Approaches to Predict In-Hospital Mortality among Neonates with Clinically Suspected Sepsis in the Neonatal Intensive Care Unit

**DOI:** 10.3390/jpm11080695

**Published:** 2021-07-22

**Authors:** Jen-Fu Hsu, Ying-Feng Chang, Hui-Jun Cheng, Chi Yang, Chun-Yuan Lin, Shih-Ming Chu, Hsuan-Rong Huang, Ming-Chou Chiang, Hsiao-Chin Wang, Ming-Horng Tsai

**Affiliations:** 1Division of Neonatology, Department of Pediatrics, Linkou Chang Gung Memorial Hospital, Taoyuan 33305, Taiwan; jeff0724@gmail.com (J.-F.H.); kz6479@cgmh.org.tw (S.-M.C.); qbonbon@gmail.com (H.-R.H.); cmc123@cgmh.org.tw (M.-C.C.); cyndi0805@yahoo.com.tw (H.-C.W.); 2School of Medicine, College of Medicine, Chang Gung University, Taoyuan 33302, Taiwan; 3Artificial Intelligence Research Center and Molecular Medicine Research Center, Chang Gung University, Taoyuan 33302, Taiwan; yfchang@mail.cgu.edu.tw (Y.-F.C.); chiyang@mail.cgu.edu.tw (C.Y.); cyulin@mail.cgu.edu.tw (C.-Y.L.); 4Department of Computer Science and Information Engineering, Providence University, Taichung 433301, Taiwan; goldfish037@gmail.com; 5Brain Research Center, National Tsing Hua University, Hsinchu 300044, Taiwan; 6Department of Computer Science and Information Engineering, Asia University, Taichung 41354, Taiwan; 7Division of Neonatology and Pediatric Hematology/Oncology, Department of Pediatrics, Chang Gung Memorial Hospital, Yunlin 61363, Taiwan

**Keywords:** neonatal mortality, artificial intelligence, big data analysis, early prediction, machine learning

## Abstract

Background: preterm and critically ill neonates often experience clinically suspected sepsis during their prolonged hospitalization in the neonatal intensive care unit (NICU), which can be the initial sign of final adverse outcomes. Therefore, we aimed to utilize machine learning approaches to predict neonatal in-hospital mortality through data-driven learning. Methods: a total of 1095 neonates who experienced clinically suspected sepsis in a tertiary-level NICU in Taiwan between August 2017 and July 2020 were enrolled. Clinically suspected sepsis was defined based on clinical features and laboratory criteria and the administration of empiric antibiotics by clinicians. The variables used for analysis included patient demographics, clinical features, laboratory data, and medications. The machine learning methods used included deep neural network (DNN), k-nearest neighbors, support vector machine, random forest, and extreme gradient boost. The performance of these models was evaluated using the area under the receiver operating characteristic curve (AUC). Results: the final in-hospital mortality of this cohort was 8.2% (90 neonates died). A total of 765 (69.8%) and 330 (30.2%) patients were randomly assigned to the training and test sets, respectively. Regarding the efficacy of the single model that most accurately predicted the outcome, DNN exhibited the greatest AUC (0.923, 95% confidence interval [CI] 0.953–0.893) and the best accuracy (95.64%, 95% CI 96.76–94.52%), Cohen’s kappa coefficient value (0.74, 95% CI 0.79–0.69) and Matthews correlation coefficient value (0.75, 95% CI 0.80–0.70). The top three most influential variables in the DNN importance matrix plot were the requirement of ventilator support at the onset of suspected sepsis, the feeding conditions, and intravascular volume expansion. The model performance was indistinguishable between the training and test sets. Conclusions: the DNN model was successfully established to predict in-hospital mortality in neonates with clinically suspected sepsis, and the machine learning algorithm is applicable for clinicians to gain insights and have better communication with families in advance.

## 1. Introduction

In premature infants or high-risk neonates hospitalized in the neonatal intensive care unit (NICU), substantial mortality may occur even though the patients overcome the most critically ill moments during the perinatal periods [1,2]. These fragile patients frequently encounter various unstable situations, some of which are due to immature organ functions, and most of which come from various invasive pathogens [3,4]. These patients are at a high risk of late-onset sepsis or clinical sepsis because of their underlying chronic comorbidities and the presence of artificial devices, as well as their inadequate immune defense and prolonged intubation and hospitalization [4,5,6]. The overall unadjusted in-hospital mortality rate for hospitalized neonates in the NICU is reported to be approximately 6.4–10.9% [1,2,3,7]. Clinically suspected sepsis, which is defined based on clinically septic features, abnormal laboratory findings and judgments of clinicians when empiric antibiotics should be used [8,9], is frequently encountered in the NICU.

During the more than three-month hospitalization course of extremely preterm neonates, optimism bias and unexpected error may lead clinicians and the parents to underestimate unstable situations of prematurity, which can delay important communication about the patient’s outcome [10,11]. Currently existing prognostic tools may aid, but they cannot be applied to all hospitalized neonates in the NICU, and inputting the data required for these tools is often time-consuming [12,13,14,15]. Recently, linear and nonlinear parameters for hospitalized patients have been modeled by machine learning (ML) algorithms with increased computational capacity [16,17]. ML algorithms using routinely collected data and electronic records have been demonstrated to predict the onset of sepsis, mortality and morbidity in order to help clinicians make more appropriate treatment decisions [16,17,18,19]. However, these ML algorithms have not been applied to neonates who are struggling with their chronic comorbidities during their hospitalization. Among various ML models, the deep neural network (DNN) method has the advantage of high interpretation, applicability, and good performance in handling big medical data, such as national network databases or international databases [17]. In this study, we aimed to develop a DNN-based multivariate regression model and validate it with six other ML algorithms that can accurately predict the in-hospital mortality of neonates with clinically suspected sepsis in the NICU.

## 2. Methods

### 2.1. Patients, Setting, and Study Design

We studied a consecutive population of all neonates hospitalized in the NICUs of Chang Gung Memorial Hospital (CGMH) between August 2017 and July 2020, in whom clinically suspected sepsis was documented during their hospitalization. The NICUs of CGMH contain a total of three units and a total capacity of 49 beds equipped with ventilator and 58 beds in special care nurseries. The annual number of admissions in these NICUs in CGMH is approximately 800 neonates. The NICUs of CGMH are the largest tertiary-level referral medical center in Taiwan and provide admission for approximately 30% of all critically ill and premature infants in Taiwan.

Clinically suspected sepsis was based on clinical and laboratory diagnosis [8,9]. The diagnostic criteria for “clinically suspected sepsis” included the following: (1) at least one of the septic symptoms, including an unstable vital sign, including fever ≥ 38 °C or hypothermia, apnea, increased oxygenation support, tachycardia or bradycardia, decreased activity, vomiting or poor intake; and (2) positive for at least two of the following experimental tests: abnormal white blood cell count (<5 × 10^6^/L or > 20 × 10^6^/L), immature to total neutrophil ratio ≥ 0.1, platelet count ≤ 100 × 10^6^/L, hemoglobin level ≤ 11.0 g/dL, C-reactive protein level ≥ 6 mg/L. In these unstable events, a blood specimen and/or sterile-site sample were obtained for culture, empirical antibiotics were prescribed, and increased life support, including ventilator support, cardiac inotropic agents, or volume expansion, was required. We prospectively followed all neonates with clinically suspected sepsis until discharge or death. Neonates with severe congenital anomalies and those who died within the first week of life were excluded. Only the first unstable episode experienced by each patient was analyzed. The neonates were randomly divided into a training set to develop the ML models and a test set to evaluate these models. This study was approved by the Institutional Review Board of Chang Gung Memorial Board, with a waiver of informed consent (the certificate no. 201802021B0).

### 2.2. Study Variables and Data Pre-Processing

Baseline demographics, including birth weight, gestational age, gender, the mode of delivery, prenatal and perinatal history and all comorbidities of the patients were collected. Chronic comorbidities included neurological sequelae, bronchopulmonary dysplasia, congenital heart disease, cholestasis, renal function impairment, and gastrointestinal diseases. Laboratory data such as white blood cell count, hemoglobin, platelet count, blood gas analysis, C-reactive protein, electrolytes, bilirubin, and renal and hepatic function results were measured at the onset of these unstable events. The onset of clinically suspected sepsis was defined when the first septic evaluation, which included laboratory examination and cultures from all potential sterile sites, was performed. The clinical symptoms included vital signs, tachycardia or bradycardia, respiratory patterns, changes in feeding conditions, medications, underlying chronic comorbidities, and the presence of artificial devices were all recorded. The primary outcome was the NICU mortality, and the discontinuation of critical care due to family requests to transfer to other hospitals was censored.

The outcomes of patients and some non-numerical variables were transferred to the corresponding numerical data and codes. After the conversion, a new numerical dataset which has 56 columns (55 variables and the outcome of all patients) was generated for the feature selection process.

### 2.3. Scoring Function

The scoring function was used to estimate the output scores by inputting the variables and the final outcomes were predicted based on the output scores. In this study, a DNN model was designed and selected as the scoring function because it exhibits excellent nonlinear computing capability and is a relatively mature technology [17]. All enrolled neonates were randomly divided, with 70% used for the training set and 30% for the testing set. Figure 1A shows the diagram of the DNN model, which includes the input layer, three hidden layers and an output layer in this study. The neuron numbers in the three hidden layers were 128, 64 and 32, respectively. In addition, a rectified linear unit (ReLU) was selected as the activation function to avoid the vanishing gradient problem [20].

### 2.4. Feature Selection of Scoring Function

The Pearson product-moment correlation coefficients (PPMCCs) of each variable were calculated and considered as the features of the neonates [21]. The feature selection process was proceeded to test different PPMCC levels to pick and to group different variables of the scoring function. Then, the scoring function of each PPMCC level was used to calculate the output scores of the testing data. Finally, the output scores were analyzed by using the receiver operating characteristic (ROC) curve to determine which PPMCC level could achieve the best predictive ability.

### 2.5. Assay Procedure

The whole assay procedure is shown in Figure 1B. The 55 variables of patients were transformed into fully numerical data through the data preprocessing and were scored by the well-trained scoring function, such as DNN and other ML methods. Finally, if the output score of a patient was higher than the cutoff value, which was determined by ROC analysis of the scoring function, the probability of a bad outcome of the patient was truly high under the present medical data. Based on such a situation, medical personnel could seriously consider further medical treatments.

### 2.6. Statistical Analysis

Statistical analyses were performed using SPSS version 15.0 (SPSS^®®^, Chicago, IL, USA) software. Categorical and continuous variables are expressed as proportions and the medians (interquartile ranges, IQRs), respectively. Categorical variables were compared by the χ^2^ test or Fisher’s exact test; odds ratios (ORs) and 95% confidence intervals (CIs) were calculated. Continuous variables were compared by the Mann–Whitney *U*-test and the *t*-test, depending on the distributions.

#### Machine Learning Procedures

In this study, a DNN model was proposed in the beginning, and then other six ML algorithms, including k-nearest neighbors (k-NN), support vector machine (SVM), random forest (RF), extreme gradient boost (XGB), Glmnet, and regression tree algorithm (Treebag) using R software (version 4.0.3), were also used for comparison. The fine-tuning of hyper-parameters, which was optimized by the five-fold cross-validations of five individual runs, and then ML models were conducted by the R package, caret (version 6.0-86). The hyper-parameter sets of these ML algorithms were pre-defined in the caret package, including the k in the KNN model, the mtry in the RF model, the sigma and cost in the SVM model with the radial basis kernel function, etc. When each ML model was constructed, all features were preselected based on the normalized feature importance to exclude irrelevancy. Finally, the F1 score, accuracy, and AUCs were calculated on the test set to measure the performance of all these models. To calculate the accuracy and F1 score of these models, the best threshold point of the receiver operating characteristic (ROC) curve was used to determine the probability of mortality. In addition, the Cohen’s kappa coefficient and the Matthews correlation coefficient (MCC) values were calculated to compare the performances of these models. All *p* values were two-sided, and the values less than 0.05 were considered significant.

## 3. Results

During the three-year study period, a total of 2472 neonates were prospectively observed, and 1095 neonates who had experienced clinically suspected sepsis and fulfilled the inclusion criteria were enrolled and analyzed. We randomly assigned 765 (69.8%) and 330 (30.2%) patients into the training and test sets, respectively. Among those with clinically suspected sepsis, only 28.5% (*n* = 312) were blood culture-positive confirmed neonatal sepsis, and all the others were blood culture-negative clinical sepsis. All these neonates received a complete course of therapeutic antibiotics for at least 5–7 days or until subsequent negative blood culture. These clinically suspected sepsis cases occurred at 19.0 (13.0–39.0) (median (interquartile range, IQR)) days of life. A total of 101 (9.2%) events had no definite diagnosis. The final in-hospital mortality rate of this cohort was 8.2% (90 neonates died) and was compatible between the training and test sets. The training set and the test set were similar in all variables. Table 1 presents the demographics and variables of the patients.

### 3.1. Feature Selection of DNN Model

The feature selection process involved the evaluation of different PPMCC levels to select different variables of the DNN model. Five different PPMCC levels were evaluated to select five different groups of variables, which were separately used as the input of the DNN model. The outputs of the DNN model were finally analyzed though the ROC curve for each PPMCC level. Figure 2A,B show the results of feature extraction optimization. The AUCs were 94.40 ± 0.49%, 92.72 ± 2.40%, 92.32 ± 3.45%, 91.52 ± 1.56% and 85.02 ± 2.57% for PPMCC levels of higher than 0.00 (all 55 variables), 0.03 (38 variables), 0.05 (27 variables), 0.1 (15 variables) and 0.15 (eight variables), respectively. The maximum AUCs of the five individual runs of DNN model for the five PPMCC levels are shown in Figure 2B.

### 3.2. Effectiveness Evaluation of the DNN Model

Nine effectiveness evaluation indexes including the AUC, sensitivity (precision or true positive rate (TPR)), specificity, false-positive rate (FPR), false-negative rate (FNR), recall (positive predictive value (PPV)), negative predictive value (NPV), F1 score, and accuracy were used to evaluate the DNN model (Table 2). We found that a PPMCC value > 0.05 had the best performance in six of nine effectiveness evaluation indexes among the five PPMCC levels. Therefore, 27 variables selected by a level, PPMCC > 0.05 were used as the covariates to develop the DNN model and others.

### 3.3. Rank of Predictors in the Prediction Scoring Function

The 27 covariates and their weights in the DNN model (PPMCC > 0.05) are shown in Table 3. The top three most influential variables in the DNN importance matrix plot were the requirement of ventilator support at the onset of clinical suspected sepsis, the feeding conditions, and intravascular volume expansion.

### 3.4. Classification Performance of DNN Model

We calculated the output scores of the test set using the DNN model. There were 330 patients in the testing set including 28 patients who died and 302 neonates who survived until discharge. Neonates who died before discharge had a significantly higher score than those who survived (0.93 vs. 0.04, *p* < 0.001; Figure 3A). The empirical and fitted ROC curves for the DNN model using the testing dataset are 0.961 and 0.951, respectively, which indicates that this DNN model can distinguish between neonates who died and those who survived until discharge. When the cutoff value of the DNN model was 0.29, the sensitivity and specificity were 89.29% and 95.36%, respectively.

### 3.5. Comparisons of the Performance of Various Machine Learning Methods

The following machine learning methods, including k-NN, SVM, RF, XGB, Glmnet, and Treebag, with the same 27 variables as input covariates, were utilized to predict in-hospital mortality (Figure 4). Regarding the efficacy of the single model that most accurately predicted the outcome, DNN exhibited the greatest AUC (0.923, 95% confidence interval [CI] 0.953–0.893) and the best accuracy (95.64%, 95% CI 96.76–94.52%), F1 score (0.77, 95% CI 0.82–0.72), and MCC value (0.75, 95% CI 0.80–0.70). In addition, Cohen’s kappa coefficient (Kappa) was used to balance the accuracy of categories to eliminate the larger Hengda’s influence. We also found that the Kappa of the proposed DNN model was highest among the seven tested models.

## 4. Discussion

In the present study, we applied ML to predict the mortality of neonates with clinically suspected sepsis during their hospitalization in the NICU. We found that the model developed using ML algorithms has much better predictive power than the traditional neonatal severity scoring systems, which are limited by the inability to fully consider integrated variables [13,14,15]. In the NICU, an episode of clinically suspected sepsis rarely directly causes mortality but is often associated with antibiotic exposure, infectious complications, prolonged ventilation and hospitalization and sometimes surgical intervention [22,23,24]. After neonates survive the most critically ill perinatal period, the combination of multiple factors, chronic comorbidities and subsequent nosocomial infections usually contributes to ultimate in-hospital mortality [25,26]. Therefore, it is often difficult to predict the final outcomes of these high-risk neonates, and this ML-based predictive model can be applicable in clinical practice.

Using ML approaches to predict NICU outcomes has emerged as a major research trend in the past decade [16,27,28,29,30], although no study has focused on neonates with clinically suspected sepsis. In the NICU, unproven culture-negative sepsis and clinical sepsis are much more common than clinicians have previously thought, but they are scarcely investigated [31,32,33]. Therefore, we included all these events in this study to encompass all clinical sepsis and undiagnosed events. Although only half of all these events were positive blood culture confirmed sepsis, all patients received empirical antibiotics and experienced at least some adverse effects, including a reposition of the central venous catheter, temporary cessation of feeding, and disturbance of the gut microbiota. As most perinatal variables may not be sufficiently precise to predict patient outcomes [13,14,19,34,35,36], it is necessary to develop an applicable model to allow clinicians to predict outcomes during the long hospitalization course of neonates. Our DNN model has the advantage of combining all underlying characteristics, the event-specific variables, and objective data, which contribute to increased accuracy and predictive power.

Several neonatal severity scores have been developed to predict the prognosis of critically ill neonates, such as the Neonatal therapeutic intervention scoring system (NTISS), Score for Neonatal Acute Physiology II (SNAP II) and SNAPPE-II scores, and Modified Sick Neonatal Score (MSNS) [13,14,15]. These scores have demonstrated accurate prediction of mortality in the NICU (AUCs of approximately 0.86–0.91) [13,34]. However, these scores were originally designed to assess the worst clinical status found in the first 24 h after admission. Previous studies found various cutoff points of these scores with inconsistent discrimination to have the highest predictive power, which reflected the influences of different institutes, quality of care and therapeutic interventions (especially the NTISS scores) [19,34,35,36]. In addition, these scores do not consider the factors of postnatal age and nosocomial infections, which undoubtedly increase the risk of mortality [19,36,37,38]. Mesquita Ramirez et al. found that SNAP II and SNAPPE II only have moderate discrimination in predicting mortality [38]. In contrast, our DNN model incorporates all easily overlooked factors at the onset of critically ill events and overcome the problems of complex input and calculation. Based on our DNN model, the probability of mortality can be calculated immediately through the established electronic medical record system.

Other important advantages of this study are the strict comparisons between our DNN model and six ML models and the various natural measures to compare these ML methods. In the field of ML algorithms, which often automatically scrutinize huge amounts of data and classify them into hundreds of unrelated categories, the evaluation process of the classifiers and comparisons of different ML models are very important [39,40]. Accuracy has been the most widely used for decades but is limited by misclassification among classes and the lack of marginal distributions [41]. Kappa is used to measure interrater reliability for qualitative items and to evaluate the agreement between the actual and assigned classes [39]. As all these measures have limitations, we applied the AUC, accuracy, F1 score, Kappa and MCC to compare all ML models. Among seven models, our DNN model is convincing given the highest AUC, accuracy, F1 score, Kappa and MCC.

There are some limitations in the current study. The application of this DNN model needs further external validation with data collected from other institutes or other countries since this is a single-center study, and some bias or issues of therapeutic policies are inevitable. Second, the sample size in this study was only moderate, although it was significantly larger than those in previous studies that analyzed neonatal severity scores [27,34,37,38] and those using ML to analyze ICU data [16]. However, a standard requirement of more than 5000 data points is preferred to establish a supervised ML with acceptable performance [39]. Last, the mortality of this study cohort was a relatively low percentage of only 8.2%, and the relative risk of the most important variables cannot be obtained.

## 5. Conclusions

We successfully applied the ML approaches to predict final in-hospital mortality when a high-risk neonate experiences clinically suspected sepsis in the NICU. We demonstrated that some important features and parameters are important for final outcome prediction. Further studies are warranted to investigate the real-time adjustment of mortality risk to optimize treatment and improve the outcomes in the NICU.

## Figures and Tables

**Figure 1 jpm-11-00695-f001:**
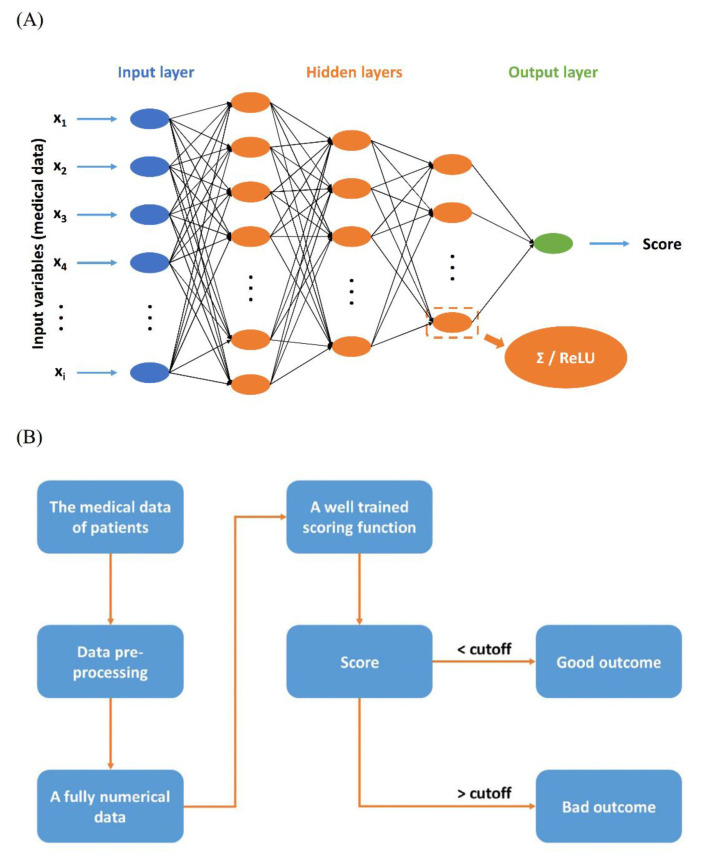
(**A**) Diagram of the deep neural networks multivariate regression model. (**B**) Diagram of the whole assay procedure. (ReLU: a rectified linear unit).

**Figure 2 jpm-11-00695-f002:**
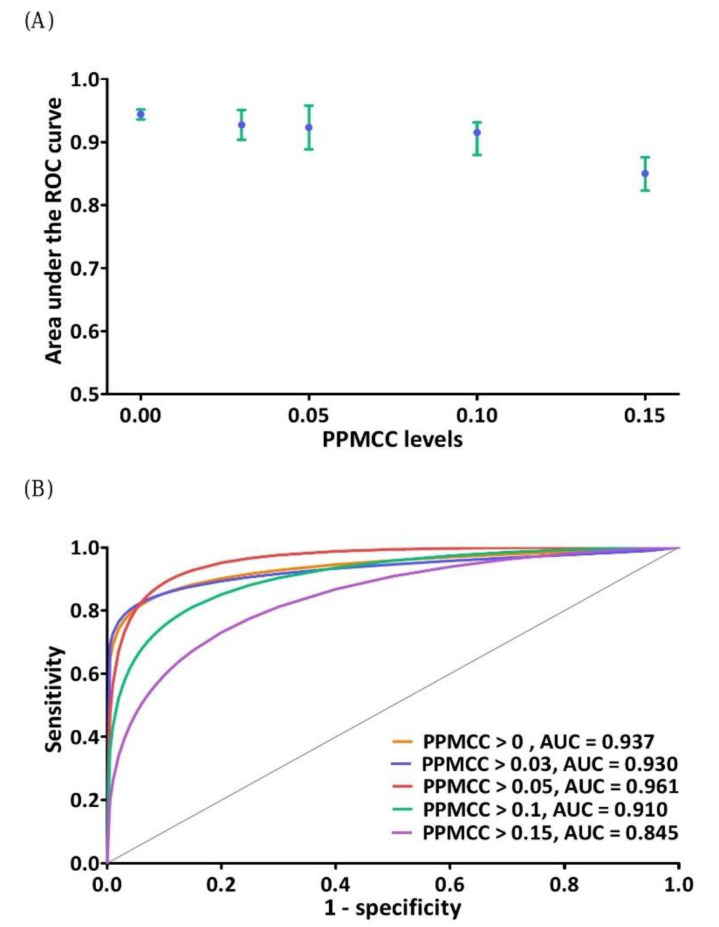
(**A**) Optimization of feature extraction. The feature extraction process was optimized by testing different PPMCC levels to select different medical data as the variables of the scoring function. The tested PPMCC levels were higher than 0.00, 0.03, 0.05, 0.1 and 0.15 corresponding to areas under the ROC curves (AUCs) of 94.40 ± 0.49%, 92.72 ± 2.40%, 92.32 ± 3.45%, 91.52 ± 1.56% and 85.02 ± 2.57%, respectively. The error indicates one standard deviation, which is the result of five individual runs of each scoring function. (**B**) The maximum AUCs of the five individual runs of each scoring function.PPMCC: The Pearson product-moment correlation coefficient; ROC: receiver operating characteristic.

**Figure 3 jpm-11-00695-f003:**
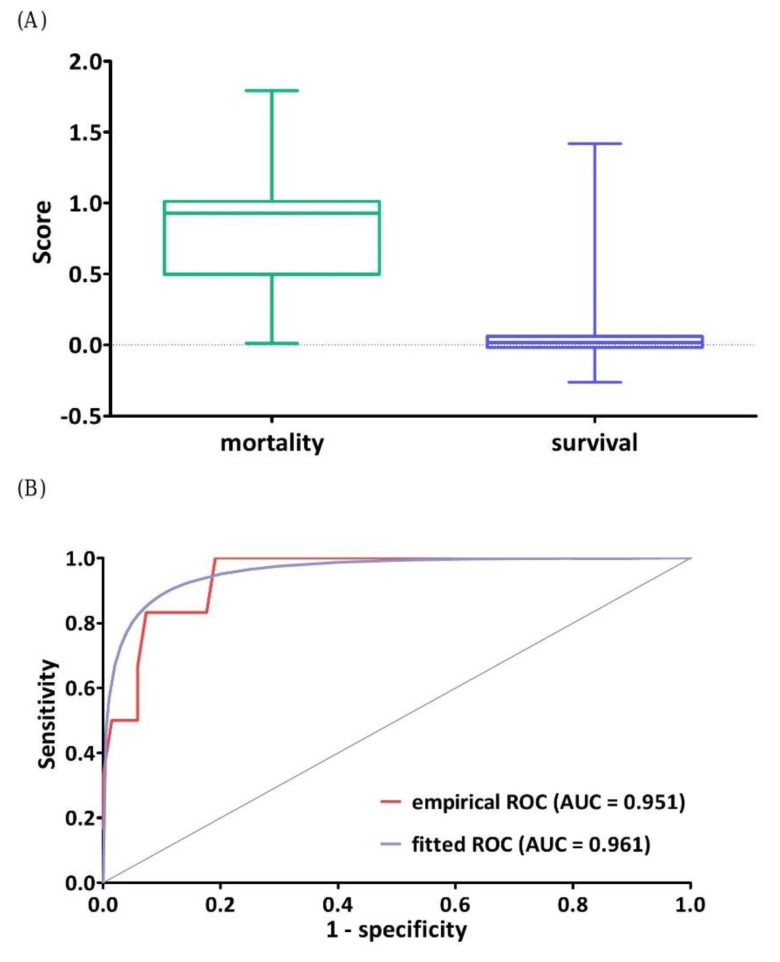
(**A**) Scores of the final survivors (*n* = 302) were compared to those who died before discharge (*n* = 28). (**B**) ROC curve analyses of the ability of the score to discriminate between the death group and the control group. ROC: receiver operating characteristic.

**Figure 4 jpm-11-00695-f004:**
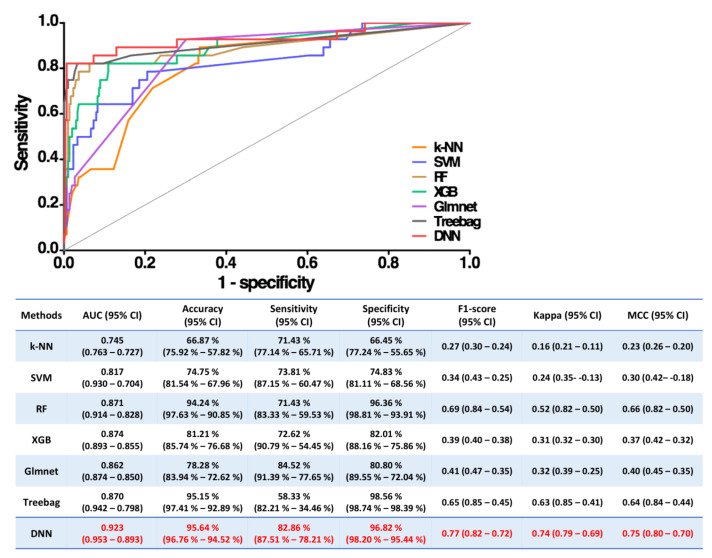
Comparison of AUCs among machine learning models. DNN yielded the greatest AUC for single-model prediction. The bar graph indicates the median value of the AUC of each model. K-NN: k-nearest neighbors algorithm; RF: random forest; SVM: support vector machine; DNN: deep neural networks; XGB: eXtreme gradient boosting; Treebag: regression tree algorithm; red color indicates the best performance.

**Table 1 jpm-11-00695-t001:** Patient demographics, characteristics, and clinical presentation of all neonates with suspected sepsis.

Characteristics	All Study Subjects (Total *n* = 1095)	The Training Set (Total *n* = 765)	The Test Set (Total *n* = 330)	*p* Values
Cases demographics				
Gestational age (weeks), median (IQR)	30.0 (27.0–35.0)	30.0 (27.0–35.0)	31.0 (27.8–36.0)	0.338
Birth weight (g), median (IQR)	1360 (1020–2150.0)	1355.0 (1040.0–2020.0)	1365 (897.5–2500.0)	0.758
Gender (male/female), n (%)	580 (53.0)/515 (47.0)	406 (53.1)/359 (46.9)	174 (52.7)/156 (47.3)	0.947
5 min Apgar score ≤ 7, n (%)	219 (20.0)	151 (19.7)	68 (20.6)	0.742
Inborn/outborn, n (%)	873 (79.9)/222 (20.1)	606 (79.2)/159 (20.8)	267 (80.9)/63 (19.1)	0.567
Birth by NSD/Cesarean section, n (%)	358 (32.7)/736 (67.3)	256 (33.5)/508 (66.5)	102 (30.9)/228 (69.1)	0.440
Respiratory distress syndrome (≥Gr II), n (%)	467 (42.6)	325 (42.5)	142 (43.0)	0.894
Perinatal asphyxia, n (%)	84 (7.7)	55 (7.2)	29 (8.9)	0.387
Underlying Chronic Comorbidities, n (%)				
Neurological sequelae	202 (18.4)	146 (19.1)	56 (17.0)	0.445
Bronchopulmonary dysplasia	189 (17.3)	139 (18.2)	50 (15.2)	0.257
Complicated cardiovascular diseases	27 (2.5)	23 (3.0)	4 (1.2)	0.091
Symptomatic patent ductus arteriosus	356 (32.5)	258 (33.7)	98 (29.7)	0.206
Gastrointestinal sequelae	55 (5.0)	34 (4.4)	21 (6.4)	0.227
Renal disorders	8 (0.7)	6 (0.8)	2 (0.6)	0.753
Congenital anomalies	36 (6.5)	29 (8.4)	7 (5.6)	0.196
Presences of any chronic comorbidities	365 (33.3)	266 (34.8)	99 (30.0)	0.142
Day of life at onset of suspected sepsis (day) ^#^	19.0 (13.0–39.0)	19.0 (14.0–37.0)	20.0 (10.0–41.3)	0.232
Previous antibiotic exposure, n (%)	662 (60.5)	472 (61.7)	190 (57.6)	0.202
Use of TPN and/or intrafat, n (%)	527 (48.1)	367 (47.8)	160 (48.5)	0.895
Use of central venous catheter, n (%)	741 (67.7)	521 (68.1)	220 (66.7)	0.673
Feeding situation *, n (%)				0.186
Nothing by mouth (NPO)	700 (63.9)	479 (62.6)	211 (63.9)	
Half amount	353 (32.2)	259 (33.9)	94 (28.5)	
Full amount	42 (3.8)	27 (3.5)	15 (4.5)	
Ventilator requirement *, n (%)				0.123
O2 hood	260 (23.7)	197 (25.7)	63 (19.1)	
Nasal cannula	45 (4.1)	34 (4.4)	11 (3.3)	
Nasal continuous positive airway pressure	151 (13.8)	90 (11.8)	61 (18.5)	
Nasal intermittent mandatory ventilation	174 (15.9)	124 (11.3)	50 (15.2)	
Intubation with mechanical ventilation	380 (34.7)	260 (34.0)	120 (36.4)	
On high frequency oscillatory ventilation	85 (7.8)	60 (7.8)	25 (7.6)	
Clinical features *, *n* (%)				
Intravascular volume expansion	185 (16.9)	137 (17.9)	48 (14.5)	0.188
Requirement of cardiac inotropic agents	91 (8.3)	62 (8.1)	29 (8.8)	0.721
Metabolic acidosis	201 (18.4)	141 (18.4)	60 (18.2)	0.932
Coagulopathy	175 (16.0)	120 (15.7)	55 (16.7)	0.187
Requirement of blood transfusion **	627 (57.3)	445 (58.2)	182 (55.2)	0.387
Laboratory data				
Leukocytosis or leucopenia	269 (24.6)	193 (25.2)	76 (23.0)	0.491
Shift to left in WBC (immature > 20%)	65 (5.9)	48 (6.3)	17 (5.1)	0.577
Anemia (hemoglobin level < 11.5 g/dL)	717 (65.5)	501 (65.5)	216 (65.4)	0.991
Thrombocytopenia (platelet < 150,000/uL)	243 (22.2)	166 (21.7)	77 (23.3)	0.526
C-reactive protein (mg/dL), median (IQR)	11.3 (8.2–23.3)	11.4 (7.5–24.0)	11.3 (8.1–22.5)	0.456

NSD: normal spontaneous delivery; IQR: interquartile range; HFOV: high-frequency oscillatory ventilator; NTISS score: Neonatal Therapeutic Intervention Scoring System; TPN: total parenteral nutrition. * At onset of suspected sepsis; ** Including leukocyte poor red blood cell and/or platelet transfusion. ^#^ Data are median (interquartile range) and analyzed by Mann–Whitney *U*-test. All other comparisons are analyzed by χ^2^ test.

**Table 2 jpm-11-00695-t002:** Effectiveness evaluation indexes of the scoring function.

	PPMCC > 0.00	PPMCC > 0.03	PPMCC > 0.05	PPMCC > 0.10	PPMCC > 0.15
Area under ROC curve (AUC)	**94.40 ± 0.49%**	92.72 ± 2.40%	92.32 ± 3.45%	91.52 ± 1.56%	85.02 ± 2.57%
Sensitivity (true positive rate)	**87.14 ± 3.19%**	81.41 ± 2.99%	82.86 ± 5.30%	82.86 ± 1.60%	82.86 ± 4.66%
Specificity (true negative rate)	92.78 ± 3.18%	95.63 ± 2.61%	**96.82 ± 1.58%**	94.70 ± 2.12%	85.36 ± 4.14%
False-positive rate (FPR)	7.22 ± 3.18%	4.37 ± 2.61%	**3.18 ± 1.58%**	5.30 ± 2.12%	14.64 ± 4.14%
False-negative rate (FNR)	**12.86 ± 3.19%**	18.57 ± 2.99%	17.14 ± 5.30%	17.14 ± 1.60%	17.14 ± 4.66%
Positive predictive value (PPV or Recall)	52.27 ± 13.88%	65.71 ± 12.88%	**72.18 ± 11.51%**	60.29 ± 8.04%	35.23 ± 5.02%
Negative predictive value (NPV)	98.74 ± 0.27%	98.23 ± 0.26%	**98.39 ± 0.48%**	98.32 ± 0.12%	98.19 ± 0.42%
F1 score [ = 2×TPR×Recall/(TPR + Recall)]	0.67 ± 0.08%	0.72 ± 0.08%	**0.77 ± 0.06%**	0.69 ± 0.05%	0.49 ± 0.04%
Accuracy	92.30 ± 2.69%	94.42 ± 2.25%	**95.64 ± 1.28%**	93.70 ± 1.81%	85.15 ± 3.45%

PPMCC: Pearson product-moment correlation coefficients; ROC: receiver operating characteristic. Bold color indicates the best performance of the effectiveness index.

**Table 3 jpm-11-00695-t003:** The weights and importance of each covariate in the development of the final predictive model.

Variables	Weights	Variables	Weights
Requirement of ventilator support	0.6016	Birth body weight (g)	0.0735
Feeding conditions	0.3938	Bronchopulmonary dysplasia	0.0628
Intravascular volume expansion	0.3623	PaO_2_ (mmHg)	0.0605
Shift to left in white blood cell (immature > 20%)	0.2397	Blood transfusion with fresh frozen plasma	0.0493
Requirement of cardiac inotropic agents	0.2258	PH value	0.0385
Peak end expiratory pressure	0.2201	Use of diuretics/bronchodilator	0.0333
Metabolic acidosis	0.2111	C-reactive protein	0.0331
Central venous catheter	0.1785	Previous antibiotic exposure *	0.0248
Apnea	0.1411	Perinatal asphyxia	0.0204
Apgar score at 5 min	0.1402	Thrombocytopenia	0.0068
PaCO_2_ (mmHg)	0.1372	Gender	0.0057
Presences of any chronic comorbidities	0.1358	On corticosteroid	0.0029
Previous positive sputum cultures *	0.1262	Fever	0.0022
Red blood cell count	0.1008		

* within one month before the life-threatening events.

## Data Availability

The datasets used/or analyzed during the current study are available from the corresponding author on reasonable request.

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
