# Peer review of "Machine Learning Approaches to Predict In-Hospital Mortality among Neonates with Clinically Suspected Sepsis in the Neonatal Intensive Care Unit"

_jpm, 2021, doi:10.3390/jpm11080695_

Round 1

Reviewer 1 Report

Overall, it is an interesting paper on the use of machine learning to predict in hospital mortality among neonates with sepsis. It compares various prediction models of machine learning. The results of this paper could led to improved patient outcomes, if adopted.

Line 72-76: What country are these statistics from? Could you insert the country into this sentence?

Line 81: Could the authors explain why the currently existing prognostic tools cannot be applied to all hospitalized neonates in the NICU?

Line 91: What does “big medical data” mean? Could the authors clarify what this means in the sentence?

Line 92: Change “other six” to “six other”.

Line 98: Where is “Chang Gung Hospital”? could the authors insert the country where this hospital is located?

Line 101: Change “annual admission” to “ annual number of admissions”

Line 104: Where do the remaining 70% of neonates go to hospital?

Line 114: How were the subjects randomly divided between the training and test set?

Line 140-143: Insert a reference for this statement

Figure 1: Missing the “e” at the end of the word outcome.

Line 170: Do the authors really mean version 15 of SPSS? This version is very old.

Line 196-198: The results show that 1095/2472 neonates had sepsis. Could the authors compare this rate with other countries to assess whether this is similar or different.

Line 200: Change “Among these suspected sepsis” to “Among those with suspected sepsis”.

Table 1: Insert a footnote to describe what statistical test the p-value is from.

Table 1: Why use median and IQR to describe the distribution of birth weight? Was the distribution skewed?

Table 1: What does “inborn” and “outborn” mean?

Table 1: What is “half amount” and “full amount”?

Line 314: Use “advantage” not “advantages”

L349: Use “sample size” instead of “the patient number”

Author Response

Dear reviewer:

       I appreciate your reviews, constructive advice and opinions. I have revised most of the comments. Please see the attachment, thank you.

Best regard,

Tsai Ming Horng

Reviewer 2 Report

Overall the article was very unclear in the main components of the hypothesis, methodology, reporting of outcomes and discussion. Insufficient information in this article does not give me confidence in the research methods employed and reported particularly how the authors conducted their result without golden criteria to define the neonates with suspected sepsis.

In intududuction section  authors referred to that: “…. clinically suspected

sepsis, which is defined when septic evaluation is done and empirical antibiotics are

prescribed, is often the first life-threatening event after the neonates survive the most

critical perinatal period” this is not true, sepsis is is often the first life-threatening event after the neonates survive the most ritical perinatal period

 In result section the mention that: “We randomly assigned 765

(69.8%) and 330 (30.2%) patients into the training and test sets, respectively. Among

these suspected sepsis, the most common causes were blood culture-positive

nosocomial infections (n=312, 28.5%), followed by culture-negative clinical sepsis

(n=224, 20.5%), apnea of prematurity (n=108, 9.9%), anemia (n=88, 8.0%),

neurological sequelae (n=62, 5.7%), and bronchopulmonary dysplasia (n=58, 5.3%). Neonates who had bronchopulmonary dysplasia or anemia had and suspected sepsis? Or were included in the group with suspected sepsis due to the clinical signs if this is true then it means that there is an error in the methodology of the study. The definition criteria of suspected sepsis have not been precisely defined.

Therefore, I recommend that this paper is not suitable for publication in Journal of Personalized Medicine.

Author Response

(The authors gave the same response as above.)

Reviewer 3 Report

In their paper, Hsu and colleagues applied machine learning approaches to predict in-hospital mortality among neonates with clinically suspected sepsis in the NICU.

The authors presented a novel, modern approach and managed to develop ML algorithms that can potentially be applied in everyday clinical work.

However, I have some remarks about the study design:

1) Why did the authors select both premature and term neonates in their analysis? 
Such a generalized approach would bias the models towards preterm neonates, since the incidence of sepsis in these patients is higher than in term neonates. Besides, the etiologies significantly differ (i.e. a higher number of premature neonates develop age-related chronic diseases (NEC, BPD)), and are having higher numbers of invasive procedures, which each on their own increase the mortality.
I suggest that the authors do two separate analyses, one for preterms, and the other for term infants. If the number of term neonates is too small for the analysis, the work would benefit more from having only the premature neonates in the analysis, which would be further needed to be specified in the title.

2) The term sepsis should be defined more precisely. Did the authors mean - clinically suspected early or late-onset sepsis or sepsis in general? If the authors meant overall sepsis - the mean/median timing when sepsis should be reported in the table.

3) In Table 1. there is no need for reporting the characteristics of the training and test set since a different random number generator would result in different characteristics.

4) How did the authors handle class imbalances? The dataset is highly imbalanced with a mortality of 8.2%. Were over/undersampling or hybrid techniques applied? If not, the authors should consider applying some those techniques (i.e. SMOTE), because a balanced dataset leads to better models.

5) Why did the authors use PPMCCs for feature selection? How did the authors handle highly correlated variables? Did the authors apply any normalization/scaling/centering before training the algorithm?

6) Figure 4) should contain both sensitivities and specificities, since a highly imbalanced ML algorithm may have a high AUC, due to class imbalances (high sensitivity, and low specificity)

Author Response

(The authors gave the same response as above.)

Round 2

Reviewer 2 Report

The authors addressed concerns and recommendations raised by this reviewer. The manuscript appeared improved than the previous version

Reviewer 3 Report

Dear editor,

in this revised version, the authors have significantly improved the quality of the manuscript, and therefore I suggest that it can be published in the present form.

Best regards,

Matej Sapina